# Sasang Constitution May Act as a Risk Factor for Depressive Symptoms—A Survey for Local Residence

**DOI:** 10.3390/healthcare10081548

**Published:** 2022-08-16

**Authors:** Yunyoung Kim, Eunsu Jang

**Affiliations:** 1Department of Nursing, Andong National University, Andong 36729, Korea; 2Department of Korean Medicine, Daejeon University, Daejeon 34520, Korea

**Keywords:** Sasang constitution, depression, risk factor

## Abstract

This study investigated whether a person’s Sasang constitution (SC) could be a risk factor for depressive symptoms. We classified the SC of 653 participants into Taeeumin (TE), Soeumin (SE), and Soyangin (SY), using the KS–15 questionnaire. We also categorized them into depressive and non-depressive symptom groups using the Center for Epidemiologic Studies Depression scale questionnaire. A *t*-test and chi-square test were used to compare the general characteristics of the depressive and non-depressive symptom groups. A one-way analysis of variance compared the scores of depressive symptoms according to SC, and a multiple logistic regression obtained the odds ratios (ORs); *p* < 0.05. The depressive symptom score for SE (13.6 ± 9.97) was significantly higher than that for SY (9.2 ± 6.51) and TE (10.8 ± 7.70; *p* < 0.001; SE > TE, SY, Scheffé). SE was associated with an increased prevalence of depressive symptoms compared with SY (OR: 2.315; 95% confidence interval [CI]: 1.389–3.860, *p* < 0.01) and TE (OR: 1.660; 95% CI: 1.076–2.561, *p* < 0.05), as well as an increased prevalence of depressive symptoms compared with SY (OR: 2.907; 95% CI: 1.379–6.144, *p* < 0.01) after adjusting for average height and distribution of living with family, medication, and drinking. This study reveals that SC, particularly SE, could be significantly associated with and be considered a risk factor for depressive symptoms.

## 1. Introduction

Depression is a common disease having a wide range of effects on thinking, mood, and physical activity. It is accompanied by symptoms such as low mood, loss of energy, sadness, loss of enjoyment, sleep disturbance, loss of appetite, fatigue, and difficulty concentrating. In Korea, about 15% of people experience depression at least once in their lifetime [1]. According to the World Health Organization, depression currently ranks third among the top 10 diseases with the greatest global burden; it is expected to rank first by 2030 [2].

Depression has high global prevalence with a large impact on various functional disorders, deterioration of quality of life, and suicide. Location of residence is related to suicide, an issue that is very serious in rural areas [3,4]. In particular, Korea has the highest suicide rate among countries in the Organisation for Economic Co-operation and Development. Therefore, it is necessary to establish a mental health policy at the national level [5].

Further, regarding the neurobiochemical risk factor for depression, it was confirmed that a serotonin (5-hydroxytryptamine) system, with which various genes are associated, affects the expression of depression and anxiety, as well as suicidal ideation [6,7]. For example, for identical twins with depression, the disease concordance rate is up to 50%. This rate is influenced by both heredity and the environment. In other words, susceptibility to the onset of depression increases when environmental influences are added to genetic predisposition [8].

Sasang constitution is a Korean personalized medicine that categorizes people into four types—Taeyangin, Soyangin, Taeeumin, and Soeumin—according to their appearance, disease, and temperament, and treats them differently according to this type. From the perspective of Sasang constitutional medicine, most humans tend to be in a skewed state in terms of the seesaw balance between the visceral groups of specific pairs: the lung–liver pair and the spleen–kidney pair [9].

Based on a skewed equilibrium of these visceral group pairs, each constitution has a different appearance, body shape, and psychological and physiological characteristics, which originate from one’s mental temperament [10]. Therefore, numerous studies have been conducted on Sasang constitution based on the hypothesis that individuals genetically inherit their Sasang constitution without change [11,12]. Many of them insist that susceptibility to disease is different for each constitution type in terms of physical aspects such as diabetes, impaired glucose tolerance, metabolic syndrome, obesity, hypertension, and pre-hypertension. Recently, some studies have suggested that some Sasang constitution types may have a tendency toward mental illnesses such as anxiety and post-traumatic stress disorder, which can be identified through simple neuro-diagnostic tests [13,14,15,16,17]. However, research on susceptibility to mental issues is still insufficient compared to that on physical issues [18].

As the foundation of Sasang constitution is based on mental traits, understanding the psychological state is considered a very important task. Accordingly, this study determines whether there is a difference in depressive tendencies based on local residents’ Sasang constitution, and whether Sasang constitution is a risk factor for such tendencies.

## 2. Materials and Methods

### 2.1. Participants and Data Collection

The data were collected by convenience sampling of local residents at a community center in a city in South Korea from October to December 2020. The researcher introduced and explained the aim of this study to the people who visited the center. Some people who were interested in the study joined. However, we excluded several individuals: people who could not understand the contents of study, people who had severe physical/psychological conditions, and people who were under 20 years old were excluded.

To protect the subjects’ rights, the study was carried out following approval by the Institutional Review Board (approval number: 1040191-202108-HR-018-01).

The number of subjects was calculated using a program for statistical power analysis (G*Power 3.1.9, Düsseldorf, Germany). Moreover, the odds ratio (OR) = 1.4, 1 − β = 0.95 and significance level (α) = 0.05 were calculated for logistic regression analysis. Accordingly, the minimum number of samples was calculated as 603. However, a sample size of 700 was determined in anticipation of the study subjects’ dropout rate.

The researcher fully explained the study’s purpose to the subjects, and after obtaining informed consent, asked them to fill in a questionnaire. Finally, 695 questionnaires were collected, and 653 were used for analysis after excluding the missing data.

### 2.2. Measurements

#### 2.2.1. Sasang Constitutional Diagnosis

The KS-15 questionnaire developed by Baek et al. was used to diagnose the subjects’ Sasang constitution [17]. It can be applied to people aged 8–80, and covers factors such as body shape, psychological traits, and usual and pathological symptoms. However, due to the low proportion of Taeyangin in the general population [19], this questionnaire was not able to classify Taeyangin. In a previous study, the constitutional diagnostic result for test–retest reliability was 87.13% (Kappa = 0.794).

#### 2.2.2. Depressive Symptom

The Center for Epidemiological Studies Depression (CES-D) scale is a measurement tool developed by the American Institute of Mental Health for the epidemiological study of depressive symptoms in the general population. The Korean version of the CES-D scale, consisting of 20 items, was corrected and supplemented by Cho and Kim to measure the degree of depression in Korean subjects [20]. The CES-D is a self-reported questionnaire that measures depressive symptoms experienced over the past week. For each response item, the score ranges from 0 to 3: 0 = extremely rare, 1 = sometimes, 2 = often, and 3 = mostly so. The total score ranges from 0 to 60, with higher scores indicating higher levels of depressive symptom. In Cho and Kim’s study, internal consistency was measured by Cronbach’s α = 0.89; in this study, Cronbach’s α = 0.725.

### 2.3. Statistical Analysis

The collected data were analyzed using the SPSS 26.0 (Armonk, NY, USA) statistical software. To examine the difference in general characteristics according to the subjects’ depressive symptoms, the cut-off score of depressive symptoms was calculated using the CES-D scale as 16 points. That is, a score of 16 or more was classified as the depressive symptom group, and a score of less than 16 was classified as the normal group. To compare the depressive and normal groups, chi-square and *t*-test were performed. In addition, differences in depressive symptoms according to Sasang constitution were analyzed by an analysis of variance and post hoc test using Scheffé’s method. Finally, logistic regression was used to determine the relative risk of depressive symptoms based on Sasang constitution, and the models were adjusted for sex, religion, and living with family as the covariance variables, which showed significant differences based on general characteristics. The statistical significance level was *p* < 0.05.

## 3. Results

### 3.1. Differences in General Characteristics According to Subjects’ Depressive Symptoms

Of the total 653 subjects, 164 (25.1%) were in the depressed group and 489 (74.9%) were in the normal group. The average heights were 160.7 cm for the depressed group 162.6 cm for the normal group, indicating a statistically significant difference (t = 2.489, *p* = 0.013). There was also a statistically significant difference between living with family and living alone (χ^2^ = 3.911, *p* = 0.048). Moreover, 150 people (24.2%) in the depressed group and 471 (75.8%) in the normal group had never taken medical drugs such as for depression, anxiety, and sleep disorders; further, 14 (43.8%) of the depressed group and 18 (56.3%) of the normal group had experience of taking such drugs. The rate of drug use experience in the depressed group was statistically significantly higher than that in the non-depressed group (χ^2^ = 3.213, *p* = 0.013). Further, there was a statistically significant difference in the subjects’ drinking status according to the depression criteria (χ^2^ = 4.833, *p* = 0.028; Table 1).

### 3.2. Difference in Depressive Symptom Score According to Sasang Constitution

Consequent to examining the differences according to Sasang constitution types by identifying the depressive symptom scores, Taeeumin scored 10.8 ± 7.70, Soeumin 13.6 ± 9.97, and Soyangin 9.2 ± 6.51. Soeumin had the highest depressive symptom score, and as a result of post hoc analysis using the Scheffé test, Soeumin’s depressive symptom score was statistically significantly higher than that of Taeeumin and Soyangin (F = 12.189, *p* < 0.001; Figure 1).

### 3.3. Distribution of Depressive Symptom According to Sasang Constitution

According to the degree of depressive symptom based on the Sasang constitution, the subjects were classified into depressed and normal groups. This classification was based on the 16 points of the depressive symptom score as measured by the CES-D questionnaire, and cross-analysis was performed according to the Sasang constitution. In the case of Soeumin, 47 people (35.1%) were classified into the depressed group and 87 people (64.9%) into the normal group. For Taeeumin, 82 patients (24.62%) were in the depressed group and 252 (75.4%) in the normal group. For Soyangin, 35 patients (18.9%) were in the depressed group and 150 (81.1%) in the normal group (χ^2^ = 10.900, *p* = 0.004). Further, there was a statistically significant difference between the depressive and non-depressive groups according to Sasang constitution, as shown in Table 2.

### 3.4. Relative Risk of Depressive Symptom According to Sasang Constitution

As a result of analyzing the relative risk of depressive symptom according to Sasang constitution, the OR of Taeeumin based on Soyangin was 1.395 (0.894~2.175), which was not statistically significant. Based on Soyangin, the OR of Soyangin was significant at 2.315 (1.389~3.860; *p* = 0.001). Moreover, the OR of Soyangin analyzed based on Soyangin was significant at 2.269 (95% CI: 1.359–3.787, *p* = 0.002) in a sex-adjusted model (Model 1). In Model 2, where sex and weight were adjusted, the OR of Soeumin was significant at 2.259 (95% CI: 1.184–4.312, *p* = 0.013). Finally, in Model 3, where height, family cohabitation, drug use, and alcohol consumption were all adjusted, the OR of Soeumin was 2.907 (95% CI: 1.379–6.144, *p* = 0.005), which was a statistically significant result. The OR of Soeumin analyzed based on Taeeumin was also significant at 1.660 (95% CI: 1.076–2.561, *p* = 0.022), which was statistically significant. In Model 1, where sex was adjusted for, the OR of Soeumin analyzed based on Taeeumin was 1.590 (95% CI: 1.026–2.463, *p* = 0.038), which was statistically significant. However, there was no statistically significant difference between Model 2, which adjusted for sex and weight, and Model 3, which adjusted for sex, weight, height, family cohabitation, drug use, and alcohol consumption (Table 3).

## 4. Discussion

This study compared the general characteristics and distribution of Sasang constitution between depressive and non-depressive groups of rural residents in Korea, and we investigated whether Sasang constitution is a risk factor for depressive symptoms.

Differences were found between the two groups in terms of average height, distribution of living with family members, medication history, and drinking characteristics. Among them, regarding drinking, Lee et al. reported that drinking affects depression, and stress from drinking mediates the relation between both of them [21]. Kim et al. found that for people living with family, the quality of life was high and depressive symptoms were low, similar to the results of this study [22]. In addition, the more diseases a person suffered from, the higher that person’s depressive tendency was. Thus, medication history was also deeply related to depression [23].

There are various self-diagnostic tools to measure the degree of depression, depending on the purpose of study. In this study, as the data were collected from the general population, CES-D, which detects various aspects of depression as a short self-report questionnaire, was used. Accordingly, the depressive symptom score of Soeumin was statistically significantly higher than that of other constitution types, which meant that Soeuminn had higher depressive tendency than other Sasang constitutions. To reveal the relation between Sasang constitution and mental stress, Seo et al. examined the depressive characteristics of Sasang constitutions using the Beck depression scale. Although their results were not significant, depressive tendencies appeared in the order of Soeumin, Taeeumin, and Soyangin, similar to the order found in this study [24].

Soeumin’s temperament is unstable, introverted, and adaptable in Dongdong Susebowon [18]. It can be estimated that the instability and variability of modern society may aggravate the depressive tendencies of Soeumin. Park et al. found that Taeeumin exhibited high extroversion and neuroticism, Soyangin exhibited high extroversion but low neuroticism, and Soeumin exhibited low extroversion and high sincerity in a psychological study of Sasang constitutions [25]. In this regard, Soeumin have low external demands for high integrity, which they experience as stress, thus increasing their depressive tendencies. However, it is unclear why Soeumin scored higher on depressive symptoms than other constitutions. Thus, further research is needed on the pathology that increases depressive tendencies in Soeumin.

Further, in this study, differences based on Sasang constitution were found with respect to the distribution of the prevalence of depressive tendencies, and the distribution of Soeumin was the highest. This is natural, because Soeumin’s depressive score was highest.

As a result of using regression analysis to determine whether Sasang constitution could be a risk factor for depressive tendencies, the ORs of Soyangin and Taeeumin were significantly higher than that of Soeumin in the crude model of depressive tendency. In particular, even after adjusting for factors that differed in the depressive group such as average height, distribution of family cohabitation, drug use, and alcohol consumption, Soeumin’s depressive propensity was significantly higher than that of Soyangin. This means that Soeumin is closely related to and is a risk factor for depressive tendencies. From this perspective, Soeumin should always be mindful of the possibility of developing depressive symptoms compared with other Sasang constitutions.

In sum, this study examined disease susceptibility based on Sasang constitution. It found that depressive tendencies are related to Sasang constitution, and suggested that a specific constitution, Soeumin, could be a risk factor for depressive symptoms. While previous studies focused on differences in susceptibility to physical diseases [13], this study is meaningful as it suggests differences in susceptibility to mental symptoms based on Sasang constitution. However, it is difficult to generalize the results because this study targeted residents of certain local areas. Further, this study has a limitation in that the accuracy of the depressive symptom diagnosis is likely poor, since only a questionnaire was used to determine the participants’ depressive tendency. Finally, as the KS-15 questionnaire cannot classify Taeyangin due to the low proportion of the total population it accounts for, this study did not include it. Additional research targeting people from various regions and age groups using more accurate methods is required.

## 5. Conclusions

This study analyzed the relationship between depressive tendencies and Sasang constitution in one rural area and examined whether Sasang constitution was a risk factor for depressive tendencies. It found that Soeumin had a significantly higher score and OR for depressive symptoms than Taeeumin and Soyangin, and that the distribution of the depressive and non-depressive symptom groups differed based on their Sasang constitution. In particular, even after adjusting for the differences in demographic information between the depressive and non-depressive symptom groups, Soeumin’s depressive tendency was significantly higher than that of Soyangin. This means that Soeumin constitution types are more vulnerable to depressive symptom than Soyangin.

## Figures and Tables

**Figure 1 healthcare-10-01548-f001:**
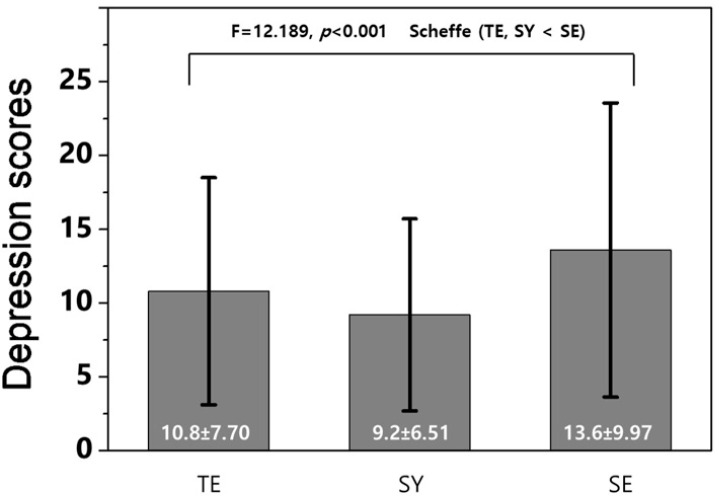
Difference in depressive symptom scores according to Sasang constitution.

**Table 1 healthcare-10-01548-t001:** General characteristics of the participant according to depressive symptom score (*n* = 653).

Variables	Total	Depressive Symptom Score		t/χ^2^	*p*
		<16 489 (74.9)	≧16 164 (25.1)		
	M ± SD/*n* (%)				
Age	53.6 ± 20.32	53.3 ± 19.75	54.8 ± 21.9	−0.767	0.444
Height	162.1 ± 8.45	162.6 ± 8.26	160.7 ± 8.86	2.489	0.013
Weight	62.1 ± 11.36	62.5 ± 11.04	60.7 ± 12.20	1.770	0.077
Gender				2.697	0.101
Male	209 (32.0)	165 (78.9)	44 (21.1)		
Female	444 (68.0)	324 (73.0)	120 (27.0)		
Occupation				0.089	0.766
No	308 (47.2)	229 (74.4)	79 (25.6)		
Yes	345 (52.8)	260 (75.4)	85 (24.6)		
Live with Family				3.911	0.048
Live together	473 (72.4)	364 (77.0)	109 (23.0)		
Live alone	180 (27.6)	125 (69.4)	55 (30.6)		
Medication				3.213	0.013
None	621 (95.1)	471 (75.8)	150 (24.2)		
Taking or Took in the past	32 (4.9)	18 (56.3)	14 (43.8)		
Religion				0.039	0.844
Have	287 (44.0)	216 (75.3)	71 (24.7)		
Do not have	366 (56.0)	273 (74.6)	93 (25.4)		
Drinking				4.833	0.028
No	453 (69.4)	328 (72.4)	125 (27.6)		
Yes	200 (30.6)	161 (80.5)	39 (19.5)		
Smoking				1.508	0.219
No	593 (90.8)	448 (75.5)	145 (24.5)		
Yes	60 (9.2)	41 (68.3)	19 (31.7)		

**Table 2 healthcare-10-01548-t002:** The distribution of depressive symptom high and low risk group according to Sasang constitution. (*n* = 653).

Variables	Depressive Symptom Score		χ^2^ (*p*)
	<16	≧16	
	*n* (%)	*n* (%)	
TE	252 (75.4)	82 (24.6)	10.900 (0.004)
SE	87 (64.9)	47 (35.1)	
SY	150 (81.1)	35 (18.9)	

**Table 3 healthcare-10-01548-t003:** Adjusted ORs (95% CI) by Sasang constitution for depressive symptom (*n* = 653).

	Depressive Symptom					
Crude	ORs (CI)					
	SY vs. TE		SY vs. SE		TE vs. SE	
	1	1.395 (0.894~2.175)	1	2.315 ** (1.389~3.860)	1	1.660 * (1.076~2.561)
Model 1	1	1.427 (0.913~2.229)	1	2.269 ** (1.359~3.787)	1	1.590 * (1.026~2.463)
Model 2	1	1.496 (0.814~2.751)	1	2.259 ** (1.184~4.312)	1	1.510 (0.740~3.080)
Model 3	1	1.184 (0.579~2.423)	1	2.907 ** (1.379~6.144)	1	2.397 (0.945~6.082)

* *p* < 0.05, ** *p* < 0.01. Model 1 = Gender; Model 2 = Gender, Weight; Model 3 = Gender, Weight, Height, Live with Family, Medication, Drinking.

## Data Availability

The data presented in this study are available on request from the corresponding author. The data are not publicly available due to privacy or ethical restrictions.

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
