# Peer review of "Sasang Constitution May Act as a Risk Factor for Depressive Symptoms—A Survey for Local Residence"

_healthcare, 2022, doi:10.3390/healthcare10081548_

Round 1

Reviewer 1 Report

This is a good paper that describes the risk factor affecting depression. It is well written. 

The abstract should be rewritten. In the results: The vertical axis of figure 1 needs a title. Also, it is unclear whether the p-value belongs to which of the bars and comparisons. the significant value.

The manuscript needs to correct the grammatical, stylistic, and grammatical errors.

Author Response

We thank the reviewer for his/her comments on our manuscript. Based on the reviewer’s comments, we have now made a few changes to the manuscript, and the response is attached with a separate file.

Reviewer 2 Report

Thank you for the opportunity to review this manuscript.    

The authors stated that “Data were collected from October to December 2020 from local residents living in 69 Area A”.

The authors need to clarify:  how the recruitment procedures of the participants were carried out?  

-         As indicated in the Methods section “The total score ranges from 0 to 60, with higher scores indicating higher levels of depression”.

The range of the scale 0-60 is long, are there categories in the scale to classify different levels of depression (mild, moderate, and severe)?  This would help to determine the level of depression, especially since the (CES-D) scale is the only subjective measure being used to assess depression. This is a major limitation of the study.

-         As shown in the statistical analysis “A score of 16 or more was classified as the depressive symptom group, and a score of less than 16 was classified as the normal group”.

In the results section, the authors stated, "Consequent to examining the differences according to Sasang constitution types by identifying the depression scores, Taeeumin scored 10.8±7.70, Soeumin 13.6±9.97, and Soyangin 9.2±6.51. Soeumin had the highest depression score”

According to the scoring system, I could understand that the mean depressive score for all groups is less than 16 which is classified as the normal group without depressive symptoms. Findings that need more clarifications and discussion to support the conclusion. 

Author Response

(The authors gave the same response as above.)

Round 2

Reviewer 2 Report

Thank you for taking my comments into your consideration.